# Metabolic Syndrome and Reproduction

**DOI:** 10.3390/ijms22041988

**Published:** 2021-02-17

**Authors:** Francesco Lotti, Sara Marchiani, Giovanni Corona, Mario Maggi

**Affiliations:** 1Andrology, Female Endocrinology and Gender Incongruence Unit, Department of Experimental, Clinical and Biomedical Sciences, University of Florence, 50139 Florence, Italy; francesco.lotti@unifi.it (F.L.); sara.marchiani@unifi.it (S.M.); 2Endocrinology Unit, Medical Department, Maggiore-Bellaria Hospital, Azienda-Usl Bologna, 40139 Bologna, Italy; 3Endocrinology Unit, Department of Experimental, Clinical and Biomedical Sciences, University of Florence, 50139 Florence, Italy; mario.maggi@unifi.it

**Keywords:** metabolic syndrome, male infertility, infertile and fertile men, semen parameters, sperm parameters, sperm DNA fragmentation, hypogonadism, testosterone, gonadotropins, treatment

## Abstract

Metabolic syndrome (MetS) and infertility are two afflictions with a high prevalence in the general population. MetS is a global health problem increasing worldwide, while infertility affects up to 12% of men. Despite the high prevalence of these conditions, the possible impact of MetS on male fertility has been investigated by a few authors only in the last decade. In addition, underlying mechanism(s) connecting the two conditions have been investigated in few preclinical studies. The aim of this review is to summarize and critically discuss available clinical and preclinical studies on the role of MetS (and its treatment) in male fertility. An extensive Medline search was performed identifying studies in the English language. While several studies support an association between MetS and hypogonadism, contrasting results have been reported on the relationship between MetS and semen parameters/male infertility, and the available studies considered heterogeneous MetS definitions and populations. So far, only two meta-analyses in clinical and preclinical studies, respectively, evaluated this topic, reporting a negative association between MetS and sperm parameters, testosterone and FSH levels, advocating, however, larger prospective investigations. In conclusion, a possible negative impact of MetS on male reproductive potential was reported; however, larger studies are needed.

## 1. Introduction

Metabolic syndrome (MetS) represents a cluster of abnormalities, including abdominal obesity, impaired glucose metabolism, hypertriglyceridemia, low HDL cholesterol and hypertension, which identifies subjects at high risk of developing type 2 diabetes mellitus (T2DM) and cardiovascular diseases (CVD) [1,2,3]. The prevalence of MetS worldwide varies greatly, ranging from 8% to 67% [1,3,4,5]. This high variation depends upon a combination of genetic, biological (e.g., age, gender, ethnicity) and social factors (e.g., urbanization, education level, socioeconomic status) as well as on the lack of globally accepted criteria defining MetS (see below) [1,3,4,5]. Despite the aforementioned considerations, the prevalence of MetS is high and is increasing worldwide [1,2,3].

MetS was first described by Reaven in his 1988 Banting lecture as “Syndrome X” [6]. Reaven suggested that insulin resistance, clustered together with glucose intolerance, dyslipidemia, and hypertension, was the main factor underlying an increased risk of CVD. The initial definition of MetS included hyperinsulinemia, impaired glucose tolerance, hypertriglyceridemia, and low HDL cholesterol. Hyperuricemia, microvascular angina, and elevated plasminogen activator inhibitor 1 were later proposed as possible additional components of the same syndrome [6,7]. Conversely, obesity was not included as part of Syndrome X, as Reaven believed that insulin resistance, instead of obesity, was the common denominator.

Several other MetS definitions have since been published [3]; however, the specific contribution of MetS to cardiovascular (CV) risk stratification is still the objective of an intense debate [3,8,9,10,11,12]. In particular, the presence of MetS seems not to offer any advantage when compared to traditional CV risk factors in predicting CV mortality and morbidities or the incidence of glucose abnormalities and T2DM [3,9,10,11,12].

A large body of evidence has clearly documented that subjects with erectile dysfunction (ED) represent a population enriched with metabolic abnormalities [13] and at a high risk of developing CV events [14,15,16]. However, even in this population the specific role played by MetS on metabolic and CV risk stratification is conflicting [17].

Emerging evidence suggests that also male infertility can be considered an early marker of poor health [18]. Large epidemiological studies published in the last decade have documented that subjects with male infertility could be considered at higher risk of hospitalization or mortality [19,20,21,22]. The specific nature of the aforementioned associations is far from having been completely elucidated, but several mechanisms, including genetic, biological, developmental, and lifestyle factors, were proposed [18]. The contribution of MetS and its related components to male fertility have been only partially investigated. The aim of the present review is to summarize and critically discuss available clinical and preclinical evidence supporting a role of MetS in male fertility.

## 2. Methods

An extensive Medline search was performed with no restrictions regarding date of publication (i.e., from inception date until December 2020) including the following words: (“metabolic syndrome”[MeSH Terms] OR (“metabolic”[All Fields] AND “syndrome”[All Fields]) OR “metabolic syndrome”[All Fields]) AND (“infertility, male”[MeSH Terms] OR (“infertility”[All Fields] AND “male”[All Fields]) OR “male infertility”[All Fields] OR (“male”[All Fields] AND “infertility”[All Fields]) AND “models, animal”[MeSH Terms] OR (“models”[All Fields] AND “animal”[All Fields]) OR “animal models”[All Fields] OR (“animal”[All Fields] AND “models”[All Fields]) AND (“semen”[MeSH Terms] OR “semen”[All Fields] OR “semens”[All Fields]) OR (“spermatozoa”[MeSH Terms] OR “spermatozoa”[All Fields] OR “sperm”[All Fields] OR “sperms”[All Fields]) AND (“parameter”[All Fields] OR “parameters”[All Fields]) AND “gonadal steroid hormones”[MeSH Terms] OR (“gonadal”[All Fields] AND “steroid”[All Fields] AND “hormones”[All Fields]) OR “gonadal steroid hormones”[All Fields] OR (“sex”[All Fields] AND “hormones”[All Fields]) OR “sex hormones”[All Fields]). The identification of relevant studies in the English language was performed independently by all the authors.

## 3. MetS Definitions

Several MetS definitions are available. These include the following: the National Cholesterol Education Program-Third Adult Treatment Panel (NCEP-ATPIII) [23], International Diabetes Federation (IDF) [24], World Health Organization (WHO) [25], American College of Endocrinology (ACE) [26], American Heart Association/National Heart, Lung and Blood Institute (AHA/NHLBI) [27] and the common definition by IDF and AHA/NHLBI (IDF&AHA/NHLBI) [28] (Table 1). At present, any definition of MetS is arbitrary, as well as the choice of the parameters to be included among MetS components, the relative weight attributed to each component and for the diagnosis, and the thresholds for each diagnostic parameter. Hence, no MetS definition can be considered superior to any other. In fact, some authors have developed MetS diagnostic criteria to identify insulin-resistant subjects, while others have aimed at predicting clinical events, including incident T2DM and CVD [29,30]. In epidemiological studies, NCEP-ATPIII criteria [24] have been used frequently due to their simplicity. Conversely, the WHO [25] and ACE [26] definitions, requiring the presence of insulin resistance or impaired glucose tolerance, are more complicated to use. In 2005, the IDF and the AHA/NHLBI attempted to reconcile the different clinical classifications. However, they produced separate recommendations [24,27], containing differences related to waist circumference and to the role of central obesity in defining MetS, considered to be a prerequisite for diagnosis by the IDF [24]. In 2009, IDF&AHA/NHLBI produced a common definition [28] diagnosing MetS in the presence of at least three of five risk factors among central obesity, elevated fasting plasma glucose, hypertriglyceridemia, low HDL cholesterol and hypertension or related treatments. Of note, the IDF&AHA/NHLBI [28] does not consider central obesity as a prerequisite for diagnosis (as in the 2005 IDF definition [24]), but as one of five criteria, and supported population- and country-specific definitions of central obesity. In addition, considering fasting plasma glucose ≥ 100 mg/dL (5.6 mmol/L) as one of five diagnostic criteria, the IDF&AHA/NHLBI definition [28] leads to MetS diagnosis in a larger population than the NCEP-ATPIII criteria [23] (considering fasting plasma glucose ≥ 110 mg/dL [6.1 mmol/L] as one of MetS criteria), the latter resulting thus as a more selective definition.

## 4. MetS and Associated Conditions

MetS represents a recognized risk factor for T2DM and CVD [1,2,3,8]. However, several other pathological conditions are associated with MetS. These conditions include in both genders: non-alcoholic fatty liver disease (NAFLD), obstructive sleep apnea, lipodystrophy, microvascular disease [1] and cancer development and mortality [31,32]. In addition, there are other MetS-associated conditions that are gender-specific, including in women policistic ovary syndrome [1,33,34] and, in men, hypogonadism [29,35,36,37,38], erectile dysfunction [29,39,40], prostatic disorders [41,42,43,44] and psychological disturbances [45,46,47].

In 2008, Kasturi et al. [48] reviewed available studies, dealing with the possible association between MetS and male reproductive health, bringing to the attention of the scientific community this relatively new topic. The authors concluded that male infertility could represent another aberration linked to MetS [48]. However, Kasturi et al.’s analysis mainly focused on the association between altered semen parameters and/or male infertility with each single MetS component, rather than with MetS as a “diagnostic category” [48]. From 2008 onwards, several original studies investigated the association between MetS as a “diagnostic category” and semen parameters and/or male infertility. The interest in this topic has increased exponentially in the last decade [49]. Three main reasons can be considered to describe this increased interest: the increasing prevalence of male infertility (estimated as 7% in 2011, up to 12% in recent years) [50], the increasing frequency of MetS worldwide -both in Western and in developing countries [3]- and its increasing prevalence in young populations, including children, adolescents and young adults of reproductive age [51,52,53,54]. However, it is worth noting that available studies used heterogeneous MetS definitions and investigated populations with different characteristics (i.e., males of infertile couples, primary or secondary infertile men, men from the general population, healthy volunteers, and fertile men). Table 2 shows the studies reporting the MetS prevalence in infertile and fertile men and those comparing its frequency between the two groups. Table 3 shows the studies published so far investigating the relationship between MetS and semen parameters, the type of cohorts studied, the MetS definition used in each study and the results on “conventional” semen parameters. Table 4 shows the associations found in the aforementioned studies between MetS, “unconventional” semen parameters (i.e., sperm DNA fragmentation and mitochondrial membrane potential) and sexual hormones. An analytical and critical analysis of all the aforementioned studies is provided below.

## 5. MetS Prevalence in Infertile and Fertile Men

Only a few studies published so far evaluated the prevalence of MetS in fertile and infertile populations, and even less have compared MetS frequency in the two populations (Table 2 and Table 3).

### 5.1. MetS Prevalence in Infertile Men

To date, MetS prevalence in infertile men was investigated in ten studies (Table 2). Ozturk et al. [55], evaluating 104 infertile men undergoing spermatic vein ligation, reported a MetS frequency of 46%. However, the authors did not report which definition of “infertility” was used, and their MetS criteria did not fit with those proposed by any of the international societes (Table 1). Males of infertile couples, defined according to the WHO [73], were investigated in nine studies [22,47,56,57,58,59,60,61,62]. IDF&AHA/NHLBI criteria were applied in three studies [47,60,61], whereas the AHA/NHLBI definition was used in one report [62], and NCEP-ATPIII criteria were used in five reports [22,56,57,58,59]. Ferlin et al. [22] also showed a significantly higher frequency of MetS in men with a low sperm count compared to those with a normal one (8.1% vs. 6.6%, respectively). In addition, Ventimiglia et al. investigated MetS in selected infertile cohorts, reporting a prevalence of 9.6% in 1337 men with primary infertility [57] and of 12% of 167 men with secondary infertility [58]. Hence, the prevalence of MetS in infertile men ranges from 7.4% to 29% considering different MetS definitions, although an outlier frequency of 46% has been also reported [55].

Interestingly, Bungum et al. [74], evaluated data from 2572 men from the population-based Malmö Diet and Cancer Cardiovascular Cohort, using information derived from questionnaires and the Swedish Tax Agency. They reported a higher prevalence (26% vs. 22%) and risk (OR = 1.22 [95% CI 0.87 to 1.72]) of MetS in childless men by comparing 333 childless men and 1817 fathers. Furthermore, Elenkov et al. [75], evaluating data derived from the Swedish registers on a large cohort of men (*n* = 459.766) who had fathered children between 2006 and 2016, reported that male partners in couples who became parents using ICSI were at a higher risk of being treated for MetS (HR = 1.28 [95% CI: 1.01–1.58]) when compared to the non IVF/ICSI men (control group).

### 5.2. MetS Prevalence in Fertile Men

The prevalence of MetS in fertile men has been investigated in three studies. In particular, Ehala-Aleksejev and Punab [59] observed MetS (NCEP-ATPIII criteria) in 12.2% of 238 men, Dupont et al. [60] in 6.1% of 100 men (IDF&AHA/NHLBI criteria), while Lotti et al. [63] using both IDF&AHA/NHLBI and NCEP-ATPIII, reported a MetS frequency of 12.9% and 6.9%, respectively, in 248 subjects.

### 5.3. Studies Comparing MetS Prevalence in Fertile and Infertile Men

Two studies have compared MetS prevalence in fertile and infertile men. Ehala-Aleksejev and Punab [59], comparing 2642 males of infertile couples and 238 fertile men, reported a significantly higher prevalence of MetS (NCEP-ATPIII criteria) in males of infertile couples than in fertile men (17.8% vs. 12.2%, respectively). Similarly, Dupont et al. [60] evaluating infertile (*n* = 96) and fertile (*n* = 100) men under 45 years of age, reported a significant higher frequency of MetS (IDF&AHA/NHLBI criteria) in infertile than fertile men (17.9% vs. 6.1%, respectively). We here report for the first time data on the MetS prevalence in fertile and infertile men derived from the database of the Andrology Unit of the University of Florence. Evaluating a consecutive series of 613 males of infertile couples (mean age 37.0 ± 7.6 years), MetS was found in 16.2% and 9.8% of the sample according to IDF&AHA/NHLBI and NCEP-ATPIII criteria, respectively. On the other hand, investigating 115 fertile men (mean age 36.6 ± 5.3 years) from a Florence spin-off of an ultrasound study on fertile men sponsored by the European Academy of Andrology [63], MetS was observed in 8.0% and 5.4% of the cohort studied according to the aforementioned criteria, respectively. Fertile and infertile men did not differ considering age (*p* = 0.083). Comparing infertile and fertile men, MetS prevalence was significantly higher in infertile men using the IDF&AHA/NHLBI criteria, but not using the NCEP-ATPIII criteria, although a trend toward statistical significance was observed (Figure 1).

Hence, even if the aformenetioned studies and the present data suggest that MetS prevalence is higher in infertile than fertile men, the available studies are limited and further investigation is advisable.

## 6. MetS and Semen Parameters

### 6.1. Cross-Sectional Studies

Studies evaluating the association between MetS and semen parameters are reported and discussed below (see Table 3, Table 4 and Table 5). The correlations between MetS and sex hormone levels reported in these studies have also been discussed (see Table 4). An analytical description of the cross-sectional studies evaluated has been reported below (Section 6.1.1). In addition, a summary of the significant associations between MetS and each seminal and hormonal parameter investigated has been provided (see Section 6.1.2 and Section 6.1.3, respectively), and a schematic representation is reported in Table 5.

#### 6.1.1. Analytical Description of the Cross-Sectional Studies

As far as we know, there are no longitudinal studies evaluating the relationship between MetS and semen quality. All available studies are cross-sectional in nature.

In 2012 Ozturk et al. [55], in a study aimed at assessing the effect of MetS upon the success of varicocelectomy in men with infertility, compared 48 men with MetS and 56 men without MetS, reporting postoperative lower sperm count and percentage of motile spermatozoa (using WHO 1999 criteria [71]) in MetS subjects. However that study presents several limitations, including no definition of “infertility” and “sperm motility”, an arbitrary MetS definition (including hypertension, high LDL cholesterolemia, low HDL cholesterolemia, hyperglycemia, obesity, physical inactivity, and blood coagulation disorders), and the lack of comparison of preoperative semen parameters between groups.

In 2013, Lotti et al. [47], evaluating 351 males of infertile couples without genetic abnormalities, reported a component-dependent, stepwise negative association between the number of MetS components (IDF&AHA/NHLBI criteria), sperm parameters (progressive motility and normal morphology), testicular echo-texture abnormality at ultrasound and testosterone levels. In particular, MetS subjects (*n* = 27) showed significantly higher rates of secondary hypogonadism compared to those without MetS (*n* = 324), and the main MetS component associated with hypogonadism increased waist circumference. After adjusting for testosterone levels, only abnormal sperm morphology retained a significant association with MetS, suggesting that hypogonadism, more than MetS itself, was responsible for the decreased sperm progressive motility [47]. In the same study, a case-control analysis showed that subjects with MetS had a significanlty lower percentage of normal sperm morphology compared with no-MetS men, even after adjusting for confounders including testosterone levels [47]. Interestingly, the only MetS component associated with abnormal sperm morphology was hypertension [47]. This finding was supported by data deriving from a previous study [76] and confirmed a subsequent study performed by our group [56] (see below). In line with this finding, a positive association between hypertension and sperm DNA fragmentation has been reported [76]. In addition, a small pilot study previously documented that a low dosage of an ACE inhibitor treatment can improve sperm parameters in normotensive men with idiopathic oligozoospermia [77] (see below). High blood pressure has been reported as a frequent but often unrecognized condition in men with primary infertility [78]. Recently, Guo et al. [79] observed that hypertensive men had worse semen quality than the normotensive counterpart. In particular, they found that hypertensive men had lower seminal volume, sperm count and motility; however, in contrast with the aforementioned previous studies [47,56,76], no difference in normal sperm morphology was reported [79]. Nevertheless, it should be recognized that the use of antihypertensive treatments, which have been linked to seminal abnormalities, was recognized as a possible confounder [79]. More recently, a study performed on fertile men [63] reported no difference in conventional semen parameters comparing subjects with and without hypertension. Hence, the relationship between hypertension and sperm morphology needs to be confirmed in futher investigations.

In 2014, evaluating 171 males of infertile couples without genetic abnormalities, we confirmed the aforementioned negative association between MetS and normal sperm morphology comparing 22 men with MetS (NCEP-ATPIII criteria) and 149 men without [56]. Respect to our previous study [47], in this study [56] we introduced insulin levels into the adjusted models as a further covariate. In the same study [56] we also reported a positive association between the increase in number of MetS components and seminal interleukin 8 (sIL-8) levels, a marker of prostate inflammation [80,81,82], and with prostate volume and signs of inflammation evaluated with color-Doppler ultrasound [81,82,83,84], which represent other factors closely related to MetS [41,42,85].

In 2014, Leisegang et al. [64] investigated a small cohort (*n* = 54) of men from the general population with multiple ethnic backgrounds. MetS was defined according to the IDF&AHA/NHLBI criteria; however cut-off values for waist circumference varied based on the ethnic and genetic backgrounds of the subjects studied. Comparing 26 subjects with MetS and 28 without, the authors found lower sperm concentration, lower total count, lower total (but not progressive) motility and vitality in the former group, but they did not assess possible differences in sperm morphology. In addition, MetS men showed a higher percentage of spermatozoa with DNA fragmentation and disturbed mitochondrial membrane potential [64]. Finally, MetS men showed lower saliva-free testosterone and progesterone levels, the latter result suggesting that steroidogenesis cascades may be compromised [64].

A subsequent study from the same authors [65] was performed on 74 participants using the same MetS criteria reported in their previous study [64]. Comparing 42 subjects with MetS and 32 without, the authors reported results similar to those found in 2014 [64] (including lower sperm concentration, total count, total motility, and vitality in MetS men), and also lower semen volume and sperm progressive motility in MetS subjects [65]. In addition, higher levels of serum and seminal pro-inflammatory factors (TNF-α, IL-1β, IL-6 and IL-8) in the MetS group were also observed [65], supporting the concept that MetS was associated with decreased fertility and with reproductive tract inflammation. In contrast with the latter findings Pilatz et al. [67], in a well-designed case–controlled study, evaluating seminal parameters including a large number of circulating and seminal cytokines in 27 subjects with MetS (IDF criteria) and 27 healthy controls, found no differences in the semen parameters and cytokine profiles between MetS and no-MetS men. However, Pilatz et al. [67] used a different MetS definition than that used by Leisegang et al. [64,65], which limits a possible comparison. As a corollary, Pilatz et al. [67] also found that MetS men had lower testosterone and SHBG levels and higher estradiol levels than no-MetS men, but no differences in gonadotropins levels.

In 2016, Elsamanoudy et al. [66] published the first study investigating the possible molecular mechanisms by which MetS can affect male fertility. The authors evaluated 120 subjects with normal semen analysis, endocrine profile, physical examination, scrotal color-Doppler ultrasound and with unknown infertility risk factors or systemic diseases. The authors compared the semen parameters of three groups: (i) 38 fertile men with MetS (IDF&AHA/NHLBI criteria), (ii) 37 infertile men with MetS (in which the only suggested risk factor for infertility was MetS) and (iii) 45 age-matched fertile volunteers without MetS (control group). They found significantly lower sperm progressive motility, normal morphology and vitality in the infertile MetS group compared with the fertile MetS one and with the control groups. In addition, comparing fertile men with and without MetS, the former group showed a lower sperm vitality. Elsamanoudy et al. [66] also reported that sperm DNA fragmentation was higher in the infertile MetS group than in the fertile MetS one, and that both MetS groups had significantly higher rates of sperm DNA fragmentation than the control group. Moreover, seminal glucose and insulin levels were higher in the infertile MetS group than in the fertile MetS and in the control groups, with insulin levels higher in the fertile MetS group than in the control one. Finally, the authors investigated the gene expression of insulin and cell death-inducing DNA fragmentation factor-α-like effector A (CIDEA) in spermatozoa, reporting that they were significantly higher in the infertile MetS group compared to the fertile MetS one, as well as in both MetS groups compared to the control group. CIDEA is a pro-apoptotic protein [86] with a role in lipid metabolism, body weight regulation and development of metabolic disorders [87]. Sperm insulin and CIDEA gene expression, as well as seminal insulin levels and sperm DNA fragmentation, were positively associated with the seminal glucose concentration in all groups. The authors [66] concluded that MetS may affect male fertility by way of the following mechanisms: (i) at the molecular level, inducing the pro-apoptotic CIDEA, leading to sperm DNA fragmentation and insulin gene expression, and (ii) through a “spermatozoa insulin resistance”, considered to be a part of MetS-related insulin resistance, characterized by increased sperm insulin gene expression, as well as increased seminal insulin and glucose levels.

In 2016, a study specifically performed on 1337 men with primary couple’s infertility [57], comparing 128 men with MetS (NCEP-ATPIII criteria) and 1209 men without MetS, found lower total testosterone (as well as inhibin B, SHBG and AMH) levels and a higher rate of hypogonadism in the MetS group, but no difference in semen parameters and in the rate of obstructive or non-obstructive azoospermia. Conversely, the same group [58], investigating 167 men with secondary couple’s infertility, reported that patients with MetS (*n* = 20; NCEP-ATPIII criteria) showed lower semen volume, sperm concentration and normal morphology than patients without MetS (*n* = 147) and confirmed lower total testosterone (as well as inhibin B, SHBG and AMH) levels and a higher rate of hypogonadism in MetS men.

In 2018, Ehala-Aleksejev and Punab [59] evaluated the impact of MetS (NCEP-ATPIII criteria) in two groups, made up of 2642 males of infertile couples and 238 fertile men. In the infertile group, comparing 471 men with MetS and 2171 men without, no difference in semen parameters was found. A similar result was observed in the fertile group, comparing 29 men with MetS and 209 without. When the authors compared the four groups (fertile MetS men, fertile no-MetS men, infertile MetS men and infertile no-MetS men), significant differences in semen parameters were observed only between fertile and infertile subjects, irrespective of the presence or the absence of MetS. In addition, a negative association between testosterone and MetS was observed in both fertile and infertile groups, while LH (but not FSH) levels were negatively correlated with MetS in the infertile group.

In 2019, Saikia et al. [69] compared semen parameters of 50 young adult men with MetS (IDF criteria) and 30 age-matched healthy men, reporting lower semen volume, total sperm count, total and progressive motility in MetS subjects. In addition, lower total testosterone, FSH and inhibin B levels were observed in MetS men, while LH levels were not evaluated.

In 2020, four studies were published on the impact of MetS on male fertility [61,62,63,68]. Chen et al. [68] evaluated a large sample (*n* = 8395) of men from the general population. A comparison between 885 men with MetS (IDF&NHLBI criteria) and 7510 men without showed a lower total (but not progressive) sperm motility and normal morphology in MetS subjects. In addition, the authors reported an inverse relationship between MetS and total sperm motility in men with ≥ four MetS components, and a negative association between MetS and normal morphology in men with one or three MetS components. Conversely, Elfassy et al. (2020) and Le et al. [62], comparing males of infertile couples with and without MetS, and Lotti et al. [63], comparing fertile subjects with and without MetS, reported no difference in semen parameters. 

Elfassy et al. [61], by defining MetS according to the IDF&AHA/NHLBI criteria, compared 45 men with MetS and 109 without MetS. Although the authors found no difference in semen parameters (including sperm DNA fragmentation) between those with or without MetS, they reported a higher infertility duration in MetS subjects, suggesting that parameters other than those classically evaluated in the semen analysis could underlie this phenomenon. The same authors [61] also evaluated the relationship among several circulating and seminal plasma adipokines (leptin, adiponectin, resistin, chemerin, visfatin, and IL-6), MetS itself and semen parameters. The most striking result was a positive correlations observed between seminal IL-6 and sperm concentration, progressive motility, and vitality. Conversely, circulating IL-6 was negatively related to sperm quality. Moreover, while men with MetS exhibited an expected lower adiponectinemia, they displayed 2.1-fold higher adiponectin levels in seminal plasma than men without MetS. The authors concluded that seminal adipokines could be involved in modulating fertility in MetS men and that seminal IL-6 could play a beneficial role on sperm function.

Le et al. [62], by defining MetS according to the AHA/NHLBI criteria, compared 65 men with MetS and 225 without MetS. They reported no difference in semen parameters comparing the two groups and, as reported above, the authors found no association between MetS and DNA fragmentation index. However, at multivariate analysis, they observed a higher sperm DNA fragmentation index in the MetS group selecting overweight individuals.

Finally, Lotti et al. [63], evaluating 248 fertile men as a part of an ultrasound project promoted by the European Academy of Andrology [63,88], found no difference in semen and scrotal color-Doppler ultrasound parameters comparing MetS and no-MetS subjects, as a result of two different analyses defining MetS according to NCEP-ATPIII or IDF&AHA/NHLBI criteria.

#### 6.1.2. Summary of the Significant Associations Between MetS and Seminal Parameters Investigated

● Semen volume

13 of 15 studies investigated the association between MetS and semen volume (Table 3 and Table 5). Three studies [58,65,69] found a lower semen volume in men with MetS than in those without, while the rest of the studies reported no difference between MetS and no-MetS men.

● Sperm concentration

11 of 15 studies investigated the association between MetS and sperm concentration (Table 3 and Table 5). Three studies [58,64,65] found a lower sperm concentration in men with MetS than in those without, while the rest of the studies reported no difference.

● Sperm total count

14 of 15 studies investigated the association between MetS and sperm total count (Table 3 and Table 5). Four studies [55,64,65,69] found a lower sperm concentration in men with MetS than in those without, while the rest of the studies reported no difference.

● Sperm motility

Out of 15 studies, six evaluated “sperm motility” (“total” ([64,65,68,69]) or “not specified” ([55,59]) motility) (Table 3 and Table 5), and 13 evaluated “sperm progressive motility” (Table 3 and Table 5).

Overall, six studies [55,64,65,66,68,69] found a lower sperm motility (regardless of the type of motility considered) in men with MetS than in those without, while three [65,66,69] reported a lower “sperm progressive motility” in MetS men. Of note, one study [66] found a lower sperm progressive motility in infertile men with MetS than in fertile men without, but no difference comparing fertile men with and without MetS (Table 3 and Table 5). The rest of the studies (Table 3 and Table 5) reported no difference between MetS and no-MetS men.

● Sperm normal morphology

13 of 15 studies investigated the association between MetS and sperm morphology (Table 3 and Table 5). Four studies [47,56,58,68] found a lower normal morphology in men with MetS than in those without. One study [66] found a lower normal sperm morphology in infertile men with MetS than in fertile men without MetS, but no difference in sperm morphology comparing fertile men with and without MetS (Table 3 and Table 5). The rest of the studies (Table 3 and Table 5) reported no difference between men with or without MetS.

● Sperm vitality

Out of 15 studies, only five [61,62,63,64,66] investigated the association between MetS and sperm vitality. Two studies [64,66] found a lower sperm vitality in men with MetS than in those without, while three [61,62,63] reported no difference.

● Sperm DNA fragmentation

Out of 15 studies, only three [61,64,65] investigated the association between MetS and sperm DNA fragmentation (Table 4). Two studies [64,65] found a lower sperm DNA fragmentation in men with MetS, while one [61] reported no difference between men with or without MetS.

● Mitochondrial membrane potential (MMP)

Out of 15 studies, only two [64,65] investigated the association between MetS and MMP (Table 4), reporting MMP lower in men with MetS than in those without.

#### 6.1.3. Summary of the Significant Associations Between MetS and Hormonal Parameters Investigated

Out of 15 studies reported in Table 3, 11 investigated associations between MetS and hormonal parameters (Table 4 and Table 5).

● Testosterone and SHBG levels

Nine studies investigated the association between MetS and circulating testosterone levels. Seven studies found lower circulating testosterone levels in men with MetS than in those without MetS [47,57,58,59,61,67,69], while two studies [56,63] reported no difference. In addition, one study [64] reported lower saliva-free testosterone levels in MetS men compared with no-MetS men. 

Four studies [57,58,61,67] investigated the association between MetS and SHBG levels, reporting lower SHBG in men with MetS than in those without.

● LH levels

Eight studies investigated the association between MetS and LH levels [47,56,57,58,59,61,63,67], reporting no difference between men with and without MetS. However, in one study [59], lower LH levels in MetS subjects were observed in males of infertile couples, but not in fertile men.

● FSH levels

Nine studies investigated the association between MetS and FSH levels. Eight studies [47,56,57,58,59,61,63,67] reported no difference between MetS and no-MetS men, while one study [69] found lower FSH levels in men with MetS than in those without MetS.

● Inhibin B levels

Four studies investigated the association between MetS and inhibin B levels [57,58,61,69], reporting lower inhibin B levels in men with MetS than in those without.

● Prolactin, AMH and estradiol levels

No difference in prolactin levels comparing men with and without MetS has been reported by [47,57,58]. Lower AMH levels in men with MetS than in those without have been reported by two studies [57,58]. No difference in estradiol levels comparing MetS and no-MetS men has been reported by [61] and [57,58,59], while higher estradiol levels in MetS men have been reported by [67]. 

● B.Meta-analysis of clinical studies

So far, only one meta-analysis [70] has assessed the effect of MetS on semen quality as well as on circulating sex hormones. The authors [70] analyzed eleven studies, with a total of 1.731 MetS cases and 11.740 controls. Compared with controls, MetS cases had a statistically significant decrease of sperm concentration, total count, progressive motility, normal morphology, and vitality, along with an increase of sperm DNA fragmentation and abnormal mitochondrial membrane potential (Figure 2). In addition, MetS cases showed a decrease in testosterone, FSH and inhibin B levels (Figure 3). No significant difference was found in semen volume, total sperm motility (Figure 2), LH, estradiol, prolactin and AMH levels (Figure 3). The authors concluded that MetS exerts a negative impact on almost all the semen parameters and part of the circulating sex hormones, tending to be a risk factor for male infertility. However, larger prospective studies were advocated by the authors to confirm their findings.

## 7. Preclinical Studies

To understand the pathophysiology underpinning possible connection(s) between MetS and male infertility, preclinical (animal) studies are very useful. A variety of westernized, high-fat diets (HFD) were administered to different rodents (different strains of rat and mouse) and, less often, to White New Zealand rabbits in order to generate a phenotype closely resembling the human MetS phenotype [89]. However, the full correspondence between the generated obesity phenotype and the human construct of MetS was verified only in a few cases. In fact, the presence of at least three of the five components of the syndrome was not often assessed, although an increase in visceral fat–the key feature of MetS–was obtained in all models. A recent systematic review and meta-analysis on HFD and male fertility in animal models was recently published [89]. After an accurate selection process, 52 studies were scrutinized and results stratified into four main broad categories: reproductive morphology of the male genital tract, standard semen analysis traits, advanced semen analysis traits (i.e., ROS and/or DNA damage), and reproductive success [89]. Sub-analyses according to the different animal species were also available [89].

In the aforementioned meta-analysis, after adjusting for animal weight, the overall relative mass of the epididymis, seminal vesicles and testis was significantly reduced by HFD, although such a decrease was not apparent in all animal species when individually investigated [89].

In 2009, we generated a rabbit model of MetS by feeding animals a HFD (4% peanut oil and 0.5% cholesterol) for 12 weeks [90]. In this rabbit model, the presence of the MetS construct-at least three components of MetS was verified in the large majority of animals, up to 75% [91]. Interestingly, we observed an HFD-dependent decrease in epididymis [92], prostate [91], seminal vesicles [90,91,93] and testis [90,92,93] weight. Figure 4A shows the MetS-induced dose-dependent decrease in testis weight, as derived from the aforementioned studies in rabbits. Such a decrease was associated with a MetS component-dependent fall in circulating testosterone levels (Figure 4B) that was associated with a decrease in gonadotropin levels, suggesting secondary hypogonadism [90,91,94]. In fact, in the preoptic area of the hypothalamus, HFD induced an increase in inflammation along with a disrupting of the complex network of neurons controlling GnRH secretion, including KISS-1, TAC3 and prodynorphin that characterize KNDy neurons [90,91,94]. The histology of the testis and of the epididymis was not substantially affected by HFD-induced MetS [91,92,93], and the presence of all eight spermatogenic stages was documented in two studies [92,93], although the number of mature spermatozoa appeared only slightly decreased [93]. However, within the HFD testis, an increased infiltration of macrophages, as characterized by RAM11 immunopositivity, was observed, along with an increased expression of genes related to inflammation [91]. Similar results were reported in the rabbit epididymis [92]. In addition, the expression of LH receptor was significantly decreased ([91] and Figure 4C), suggesting a testicular contribution to the testosterone fall. When steroidogenesis was considered, we found that MetS induced a decrease in the expression of all the genes related to testosterone formation [90,91]. The most evident results were observed in 17β-hydroxysteroid dehydrogenase 3 (HSD17B3)-the enzyme devoted to testosterone formation from ∆4-androstenedione-with, as a final result, a fall in the ratio between testosterone and ∆4-androstenedione, as assessed by mass spectrometry of testis homogenates ([91] and Figure 4D,E).

The Crean and Senior meta-analysis demonstrated an overall significant decrease in sperm number across animal species, although this result was not statistically significant in a sub-analysis considering only rabbit studies [89]. Figure 5A, shows results concerning sperm concentration obtained in our laboratory by using the aforementioned rabbit model: a trend toward a reduction was evident, without reaching statistical significance [91,92]. Results obtained on the effect of HFD on sperm motility and morphology were more homogeneous across species, as they were significant either overall or individually considering rodent and rabbit models of HFD [89]. Figure 5B−D show results in our rabbit model, according to a previous publication [92] and unpublished observations. The most impressive results were obtained on sperm morphology (Figure 5B). In particular, all MetS components-impaired glycaemia, hypertension, dyslipidemia and increased visceral fat-significantly contribute to an altered sperm morphology in multivariate analysis (*p* < 0.05 for all). In particular, abnormal sperm morphology was dose-dependently correlated with the number of MetS components present in the rabbits examined (Figure 5C). 

Concerning advanced semen analyses, the Crean and Senior meta-analysis [89] demonstrated an overall effect of HFD in increasing ROS production and DNA damage. We were not able to confirm these rodent findings in the rabbit MetS model [92,93]. However, we did find a significant, HFD-induced, impairment in the number of progesterone-induced acrosome reactions in rabbit sperm [92], suggesting functional sperm alterations.

It is possible that all the aforementioned sperm alterations have functional consequences. In fact, the Crean and Senior meta-analysis demonstrated a significant decrease in mating and fertilization success in the rodent models, without affecting the implantation process and the litter size [89]. Hence, rodents fed an HFD are less likely to mate successfully and, more importantly, the mating induced a lower number of pregnancies. Information on rabbit mating is, at present, not available.

## 8. Treatment of MetS and Its Impact on Semen Quality

So far, no study has evaluated the possible impact of MetS treatment on semen quality. However, some studies assessed the effect on seminal characteristics of medications used to treat the single MetS components. 

### 8.1. Treatment of Impaired Glucose Metabolism and Its Impact on Semen Quality

The most studied medication used to treat impaired glucose metabolism is metformin. A positive effect of metformin on male spermatogenesis has been reported in both human and animal models. 

Studies in humans are limited [95,96,97]. Morgante et al. [95] reported that a six-month treatment of 45 oligo-terato-asthenozoospermic patients with MetS with metformin (850 mg/day for the first week, 850 mg twice a day in the second week and 850 mg three times a day for the rest of the treatment period) led to a significant improvement in sperm concentration, motility, and normal morphology. The authors suggested that the improvement of semen characteristics was associated with the metformin-related reduction of insulin resistance and SHBG levels and increase of total and free testosterone levels. Bosman et al. [96] reported that a three-month treatment of 34 hyperinsulinaemic men with metformin (starting with 500 mg/day and increasing the dose until the blood sugar was controlled), alone (*n* = 19) or associated with an antioxidant treatment (*n* = 15), led to improvement of sperm normal morphology and chromatin packaging quality. Of note, sperm chromatin condensation plays a key role in male fertility, early embryonic growth and pregnancy outcomes [98]. La Vignera et al. [97] reported that the addition of slow-release metformin (500 mg/day) to FSH treatment (150 units three times a week) in insulin-resistant patients with normogonadotropic idiopathic infertility improved the efficacy of FSH therapy on spermatogenesis. In fact, comparing the characteristics of infertile men treated for three months with FSH alone (*n* = 44) and those of men treated with FSH plus metformin (*n* = 35), the authors observed higher sperm concentration, progressive motility, normal morphology, and sperm DNA fragmentation normalization rate in the latter group. Conversely, some authors [99] reported a negative effect of metformin on human spermatozoa motility and signaling pathways.

Several studies on animal models reported that metformin ameliorates testicular function and sperm quality in male mice [100,101] and rats [102,103,104] exposed to an obesogenic (high-fat [89,105] or high-sugar [102]) diet, as well as in streptozotocin-induced diabetic rats [106,107,108]. Conversely, some authors [109] reported a negative effect of metformin in Sertoli cell proliferation and daily sperm production in rats.

Studies on anti-diabetic drugs other than metformin are limited. A case report of a 35-year-old man with primary infertility, a slight increase in glucose levels and overweight showed a deleterious effect of liraglutide on male reproductive function [110]. On the other hand, some authors [111] reported that gliclazide, alone or in combination with atorvastatin, ameliorated reproductive damage in streptozotocin-induced type 2 diabetic male rats.

### 8.2. Treatment of Hypertension and Its Impact on Semen Quality

A few studies, performed in small cohorts, investigated in humans the effect of antihypertensive drugs on semen parameters, with contradictory results. 

Yamamoto et al. [112] reported that after treating 20 idiopathically infertile men with bunazosin (α1-blocker) and procaterol (β-stimulator) for five months, an increase in sperm count and seminal volume occurred in 80% of cases. In addition, the authors reported that after treatment, three pregnancies occurred, and five of six azoospermic men of the cohort studied became oligospermic. A previous study [113] demonstrated the presence of adrenergic α- and β-receptors in the myoid cells of human seminiferous tubules, and that their stimulation resulted in myoid cells contraction and relaxation, respectively. Hence, the authors suggested that the increase in sperm output could be associated with relaxation of myoid cells, leading to dilatation of stenotic areas of the seminiferous tubules and subsequent maintenance of good tubular fluid flow [112]. In a subsequent study, Gülmez et al. [114], treating for seven days 27 infertile men with several medications (prazosin, an α1-blocker, and terbutaline, β2-stimulator) similar to those used by Yamamoto et al. [112], found no difference in sperm parameters and a decrease in semen volume compared to baseline. The authors suggested that their results, at odds with what was previously reported [112], could be related to the short duration of the treatment.

Recently, a systematic review [115] documented no effect of captopril, an ACE inhibitor, on semen quality. Conversely, a previous 5-year randomized, controlled, crossover pilot study [77], performed on 28 normotensive men with idiopathic oligospermia and infertility, reported that a low dosage of a different ACE inhibitor, lisinopril, improved sperm parameters. In particular, after treatment (crossover point at week 96 and end of the study at week 282), an increase in sperm total count, motility and normal morphology and a normalization of seminal parameters in 53.6% of the participants was observed. In addition, during the 4-year follow up of the study, a pregnancy rate of 48.5% was observed.

Regarding animal models and in vitro studies, three recent reviews [50,116,117], evaluating the impact of drugs on male fertility, reported that several antihypertensive medications (including beta-blockers, alpha-blockers, calcium channel blockers, ACE inhibitors, diuretics–spironolactone- and methyl-dopa) exert a negative impact on spermatogenesis and sperm parameters. However, a study [118], not considered in the aforementioned reviews, reported that manidipine improved spermatogenesis in stroke-prone spontaneously hypertensive rats.

Due to the contradictory results of pre-clinical and clinical studies, further large longitudinal studies are needed to elucidate the relationship between antihypertensive medications and male fertility.

### 8.3. Treatment of Dyslipidemia and Its Impact on Semen Quality

A few studies reported the effect of statins on seminal parameters in human and animal models, while no study has evaluated the impact of fibrates on semen quality in humans. Recently, a systematic review [119] documented that statins exert a strong to minimal negative effect on semen quality. Of note, the largest studies considered in the review reported a negative effect of statins on semen volume [120,121], sperm concentration [121] and motility [122]. Conversely, in animals, statins were found to ameliorate semen quality characteristics [119], especially in HFD-induced-obesity [123] and in diabetic [124] male rats. Regarding fibrates, a negative effect on reproductive function has been reported in male rats (but not in humans) by a few studies [117,125].

### 8.4. Treatment of Obesity and Its Impact on Semen Quality

The paradigm of the effect of obesity treatment on semen quality is represented by studies evaluating seminal changes after bariatric surgery. A recent review and meta-analysis [126], including a total of 28 cohort studies with 1022 patients, reported that sustained weight loss induced by bariatric surgery was associated with a significant improvement of male reproductive hormones (including increase in total and free testosterone and decrease in estradiol and PRL levels), but did not improve sperm quality and function.

In summary, the use of metformin to ameliorate the semen quality of MetS patients is supported by the few available studies. The role of antihypertensive medications is debated (possible positive or null effect on sperm parameters) and needs larger longitudinal studies. Statins seem to have a negative effect on semen characteristics, while bariatric surgery seems not to improve sperm quality and function. However, caution on this topic is needed, since available studies are limited and often performed on small cohorts. Larger longitudinal studies are therefore advocated.

## 9. Conclusions

In conclusion, while several clinical and preclinical studies strongly support an association between MetS and hypogonadism [29,35,36,37,38,127], contrasting results have been reported on the relationship between MetS and semen parameters, and available studies used heterogeneous MetS definitions and investigated heterogeneous populations. So far, only one meta-analysis [70] has evaluated this topic, reporting a negative association between MetS and sperm parameters; however, advocating larger prospective investigations. Preclinical studies (meta-analyzed in [89]) were essentially in line with the clinical ones. In addition, they suggest that a low-grade inflammation is the main mechanism underlying the negative relationship between MetS and altered semen parameters. However, whether or not MetS is able to affect the ability of fatherhood, as in the case of the female counterpart, and whether or not its treatment can ameliorate the male fertility potential, is still undetermined and investigated in a few clinical and preclinical cohorts. 

## Figures and Tables

**Figure 1 ijms-22-01988-f001:**
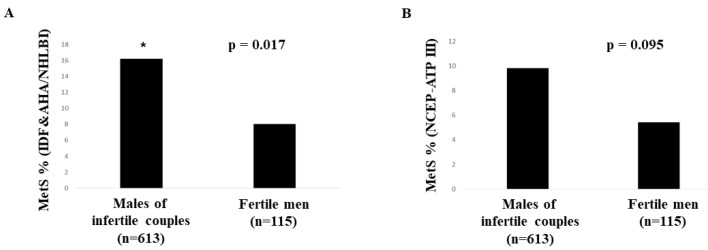
Comparison of metabolic syndrome (MetS) prevalence in fertile and infertile men derived from the database of the Andrology Unit of the University of Florence, according to IDF&AHA/NHLBI (**A**) and NCEP-ATPIII (**B**) criteria. * *p* < 0.05.

**Figure 2 ijms-22-01988-f002:**
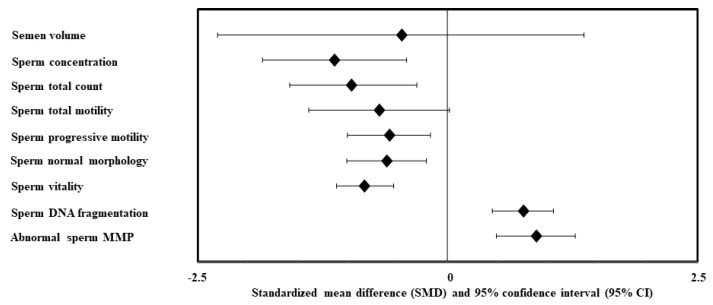
Summary of the Zhao and Pang [70] meta-analysis on the effects of metabolic syndrome (MetS) on semen parameters. MMP, mitochondrial membrane potential.

**Figure 3 ijms-22-01988-f003:**
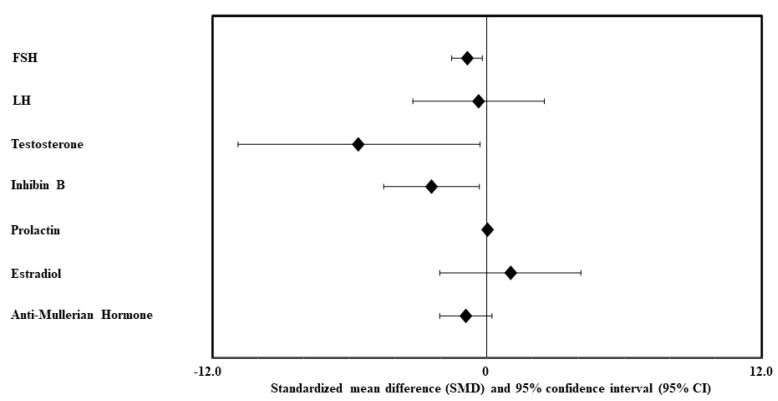
Summary of the Zhao and Pang [70] meta-analysis on the effects of metabolic syndrome (MetS) on sexual hormones.

**Figure 4 ijms-22-01988-f004:**
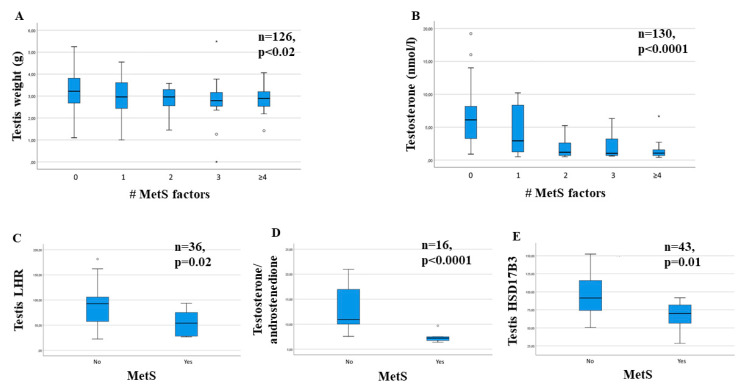
Effect of high-fat diet (HFD)-induced metabolic syndrome (MetS) on testicular weight and function. (**A**) and (**B**) show the dose-dependent effect of having the indicated numbers of MetS components on testis weight and circulating testosterone, respectively (**C**,**D**,**E**) show the effect of MetS, as a dummy variable, on testicular expression of the LH receptor (LHR), testosterone/androstenedione ratio and expression of the genes for the enzyme 17β-hydroxysteroid dehydrogenase 3 (HSD17B3), respectively. Numbers of animals examined (*n*), along with level of significance (*p* value) of the statistical analyses performed are also reported; g, grams. #, number. * and °, outlier cases.

**Figure 5 ijms-22-01988-f005:**
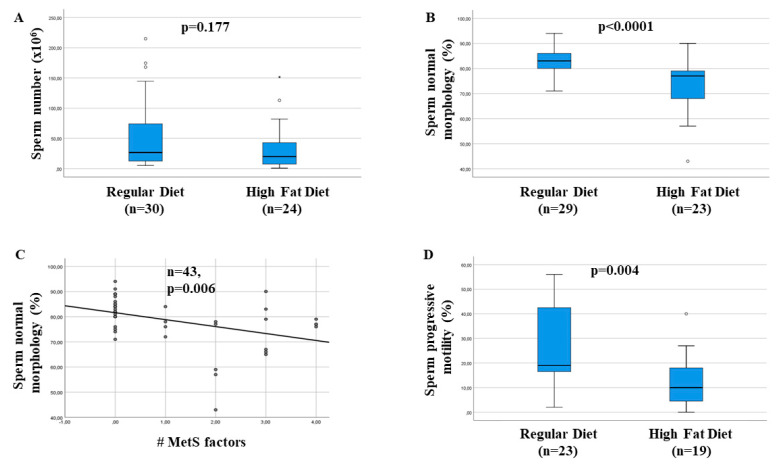
Effect of feeding rabbits a high-fat diet (HFD) on sperm number (**A**), sperm normal morphology (**B**) and sperm progressive motility (**D**). (**C**) shows the association between increasing number (#) of metabolic syndrome (MetS) components and sperm normal morphology. Numbers of animals examined (*n*), along with level of significance (*p* value) of the statistical analyses performed are also reported. * and °, outlier cases.

**Table 1 ijms-22-01988-t001:** Comparison of metabolic syndrome (MetS) definitions: National Cholesterol Education Program-Third Adult Treatment Panel (NCEP-ATPIII) and International Diabetes Federation (IDF), World Health Organization (WHO), American College of Endocrinology (ACE), American Heart Association/ National Heart, Lung and Blood Institute (AHA/NHLBI) and common definition by IDF and AHA/NHLBI (IDF&AHA/NHLBI).

NCEP-ATPIII [23]	IDF [24]	WHO [25]	ACE [26]	AHA/NHLBI [27]	IDF&AHA/NHLBI [28]
3 or more of the following	Central obesity (waist circumference ≥ 94 cm) and 2 or more of the following	Fasting insulin in top 25%; fasting glucose ≥ 100 mg/dL (6.1 mmol/L); 2 h glucose ≥ 140 mg/dL (7.8 mmol/L) and 2 or more of the following	High risk of insulin resistance: 2h plasma glucose ≥ 140 (7.8 mmol/L) and < 200 mg/dL (11 mmol/L) and 2 or more of the following	3 or more of the following	3 or more of the following
Central obesity (waist circumference >102 cm)		Obesity waist/hip ratio > 0.9 or BMI ≥ 30 kg/m^2^		Central obesity (waist circumference > 102 cm)	Central obesity (population- and country-specific definitions)
Hypertriglyceridaemia: triglycerides ≥ 150 mg/dL (1.7 mmol/L) or treatment	Hypertriglyceridaemia: triglycerides ≥ 150 mg/dL (1.7 mmol/L) or treatment	Hypertriglyceridaemia: triglycerides ≥ 150 mg/dL (1.7 mmol/L) or treatment	Hypertriglyceridaemia: triglycerides ≥ 150 mg/dL (1.7 mmol/L) or treatment	Hypertriglyceridaemia: triglycerides ≥ 150 mg/dL (1.7 mmol/L) or treatment	Hypertriglyceridaemia: triglycerides ≥ 150mg/dL (1.7 mmol/L) or treatment
Low HDL cholesterol: < 40 mg/dL (1.03 mmol/L) or treatment	Low HDL cholesterol: < 40 mg/dL (1.03 mmol/L) or treatment	Low HDL cholesterol: < 35 mg/dL (0.9 mmol/L) or treatment	Low HDL cholesterol: < 40 mg/dL (1.03 mmol/L) or treatment	Low HDL cholesterol: < 40 mg/dL (1.03 mmol/L) or treatment	Low HDL cholesterol: < 40 mg/dL (1.03 mol/L) or treatment
Hypertension: blood pressure ≥ 130/85 mmHg or treatment	Hypertension: blood pressure ≥ 130/85 mmHg or treatment	Hypertension: blood pressure ≥ 140/90 mmHg or treatment	Hypertension: blood pressure ≥ 130/85 mmHg or treatment	Hypertension: blood pressure ≥ 130/85 mmHg or treatment	Hypertension: blood pressure ≥ 130/85 mmHg or treatment
Fasting plasma glucose: **≥** 110 mg/dL (6.1 mmol/L) or diabetes	Fasting plasma glucose: ≥ 100 mg/dL (6.1 mmol/L) or diabetes	Microalbuminuria: urin albumi/urinary creatinine ratio ≥ 3.39 mg/mmol (30 mg/g)	Fasting glucose ≥ 100 mg/dL (5.6 mmol/L)	Fasting plasma glucose: **≥** 100 mg/dL (5.6 mmol/L) or treatment	Fasting plasma glucose: ≥ 100 mg/dL (5.6 mmol/L) or treatment

**Table 2 ijms-22-01988-t002:** Prevalence of metabolic syndrome (MetS) in infertile and fertile men.

Author	Cohort Studied (# Men, Country, Mean Age)	MetS Definition Used	MetS Prevalence
**Infertile Men**			
Ozturk et al., (2012) [55]	104 infertile men (Turkey) (28.2 years)	Arbitrary ^§^	46.0%
Lotti et al., (2013) [47]	351 males of infertile couples (Italy) (36.0 ± 8.0 years)	IDF & AHA/NHLBI	7.7%
Lotti et al., (2014) [56]	171 males of infertile couples (Italy) (36.6 ± 8.4 years)	NCEP-ATPIII	12.9%
Ventimiglia et al., 2016 [57]	1337 men with primary infertility (Italy) (36.5 years)	NCEP-ATPIII	9.6%
Ventimiglia et al., 2017 [58]	167 men with secondary infertility (Italy) (22−68 years)	NCEP-ATPIII	12.0%
Ehala-Aleksejev and Punab (2018) [59]	2642 males of infertile couples (Estonia) (32.6 ± 5.7 years)	NCEP-ATPIII	17.8%
Ferlin et al. (2019) [22]	5177 males of infertile couples (Italy) (31.7 ± 7.9 years)	NCEP-ATPIII	7.4%
Dupont et al. (2019) [60]	96 fertile men (France) (33.3 years)	IDF & AHA/NHLBI	17.8%
Elfassy et al. (2020) [61]	154 males of infertile couples (France) (37.1 ± 0.4 years)	IDF & AHA/NHLBI	29.0%
Le et al. (2020) [62]	290 males of infertile couples (Vietnam) (35.3 ± 5.9 years)	AHA/NHLBI	22.0%
**Fertile men**			
Ehala-Aleksejev and Punab (2018) [59]	238 fertile men (Estonia) (32.0 ± 6.1 years)	NCEP-ATPIII	12.2%
Dupont et al. (2019) [60]	100 fertile men (France) (34.4 years)	IDF & AHA/NHLBI	6.1%
Lotti et al. (2020) [63]	248 fertile men (Italy) (35.3 ± 5.9 years)	IDF & AHA/NHLBI	12.9%
		NCEP-ATPIII	6.9%
**Comparison of MetS prevalence of fertile and infertile men**			
Ehala-Aleksejev and Punab (2018) [59]	2642 males of infertile couples vs. 238 fertile men	NCEP-ATPIII	17.% vs. 12.2% (*p* = 0.028)
Dupont et al. (2019) [60]	96 infertile vs. 100 fertile men	IDF & AHA/NHLBI	17.9% vs. 6.1% (*p* = 0.012)

^§^ MetS defined as with “hypertension, high LDL cholesterolemia, low HDL cholesterolemia, hyperglycemia, obesity, physical inactivity and blood coagulation disorders” [55]. #, number.

**Table 3 ijms-22-01988-t003:** Studies investigating the relationship between MetS and “conventional” semen parameters.

Author	Cohort Studied	MetS Definition	# Subjects	Type of Analysis	Semen Volume	Sperm Concentration	Sperm Total Count	Sperm Motility	Sperm Normal Morphology	Sperm Vitality
Ozturk et al., 2012 [55] ^	Men with infertility and varicocele (Turkey)	Arbitrary ^§^	MetS, 48 No-MetS, 56	Comparison of MetS vs. no-MetS men	NE	NE	Lower in MetS men	Lower in MetS men	No difference	NE
Lotti et al., 2013 * [47]	Males of infertile couples (Italy)	IDF & AHA/NHLBI	MetS, 27 No-MetS, 324	Comparison of MetS vs. no-MetS men	No difference	No difference	No difference	No difference (p)	Lower in MetS men	NE
				Correlation between # of MetS components and seminal parameters	No correlation	No correlation	No correlation	No correlation	Negative correlation	NE
Lotti et al., 2014 [56]	Males of infertile couples (Italy)	NCEP-ATPIII	MetS, 22 No-MetS, 149	Comparison of MetS vs. no-MetS men	No difference	No difference	No difference	No difference (p)	Lower in MetS men	NE
				Correlation between # of MetS components and seminal parameters	No correlation	No correlation	No correlation	No correlation	Negative correlation	NE
Leisegang et al., 2014 * [64]	Men from the general population (South Africa)	IDF & AHA/NHLBI	MetS, 26 No-MetS, 28	Comparison of MetS vs. no-MetS men	No difference	Lower in MetS men	Lower in MetS men	Total (but not progressive) motility lower in MetS men	NE	Lower in MetS men
Leisegang et al., 2016 * [65]	Men from the general population (South Africa)	IDF & AHA/NHLBI	MetS, 42 No-MetS, 32	Comparison of MetS vs. no-MetS men	Lower in MetS men	Lower in MetS men	Lower in MetS men	Total and progressive motility lower in MetS men	NE	NE
Ventimiglia et al., 2016 * [57]	Men with primary infertility (Italy)	NCEP-ATPIII	MetS, 128 No-MetS, 1209	Comparison of MetS vs. no-MetS men	No difference	No difference	No difference	No difference (p)	No difference	NE
Elsamanoudy et al., 2016 * [66]	37 infertile men with MetS and 45 fertile men w/o MetS (Egypt)	IDF & AHA/NHLBI	MetS, 37 No-MetS, 45	Comparison of MetS vs. no-MetS men	No difference	NE	No difference	Lower in MetS men (p)	Lower in MetS men	Lower in MetS men
	38 fertile men with MetS and 45 fertile men w/o MetS (Egypt)	IDF & AHA/NHLBI	MetS, 38 No-MetS, 45	Comparison of MetS vs. no-MetS men	No difference	NE	No difference	No difference (p)	No difference	Lower in MetS men
Pilatz et al., 2017 * [67]	27 MetS men and 27 healthy men (Germany)	IDF	MetS, 27 No-MetS, 27	Comparison of MetS vs. no-MetS men	No difference	No difference	No difference	No difference (p)	No difference	NE
Ventimiglia et al., 2017 * [58]	Men with secondary infertility (Italy)	NCEP-ATPIII	MetS, 20 No-MetS, 147	Comparison of MetS vs. no-MetS men	Lower in MetS men	Lower in MetS men	No difference	No difference (p)	Lower in MetS men	NE
Ehala-Aleksejev and Punab, 2018 * [59]	Males of infertile couples (Estonia)	NCEP-ATPIII	MetS, 471 No-MetS, 2171	Comparison of MetS vs. no-MetS men	No difference	No difference	No difference	No difference	No difference	NE
	Fertile men (Estonia)	NCEP-ATPIII	MetS, 29 No-MetS, 209	Comparison of MetS vs. no-MetS men	No difference	No difference	No difference	No difference	No difference	NE
Chen et al., 2019 * [68]	Men from the general population (China)	IDF & AHA/NHLBI	MetS, 885 No-MetS, 7510	Comparison of MetS vs. no-MetS men	No difference	No difference	No difference	Total (but not progressive) motility lower in MetS men	Lower in MetS men	NE
				Correlation between # of MetS components and seminal parameters	No correlation	No correlation	No correlation	Inverse relationship with men with ≥ 4 MetS components	Inverse relationship with men with 3 MetS components	NE
Saikia et al., 2019 * [69]	50 Young adult males with MetS and 30 age-matched healthy males (India)	IDF	MetS, 50 No-MetS, 30	Comparison of MetS vs. no-MetS men	Lower in MetS men	NE	Lower in MetS men	Total and progressive motility lower in MetS men	No difference	NE
Elfassy et al., 2020 [61]	Males of infertile couples (France)	IDF & AHA/NHLBI	MetS, 45 No-MetS, 109	Comparison of MetS vs. no-MetS men	No difference	No difference	No difference	No difference (p)	No difference	No difference
Le et al., 2020 [62]	Males of infertile couples (Vietnam)	AHA/NHLBI	MetS, 65 No-MetS, 225	Comparison of MetS vs. no-MetS men	NE	No difference	NE	No difference (p)	No difference	No difference
Zhao and Pang, 2020 [70]	Meta-analysis	Various	MetS, 1731 No-MetS, 11740	Comparison of MetS vs. no-MetS men	Lower in MetS men	Lower in MetS men	Lower in MetS men	Lower in MetS men	Lower in MetS men	Lower in MetS men
Lotti et al., 2020 [63]	248 fertile men (Italy)	IDF & AHA/NHLBI	MetS, 32 No-MetS, 216	Comparison of MetS vs. no-MetS men	No difference	No difference	No difference	No difference (p)	No difference	No difference
		NCEP-ATPIII	MetS, 17 No-MetS, 231	Comparison of MetS vs. no-MetS men	No difference	No difference	No difference	No difference (p)	No difference	No difference

NE, not evaluated; w/o, without. * Studies included in Zhao and Pang meta-analysis [70]. ^§^ MetS defined as with “hypertension, high LDL cholesterolemia, low HDL cholesterolemia, hyperglycemia, obesity, physical inactivity and blood coagulation disorders” [55]. ^ Semen analysis performed according to WHO 1999 criteria [71]. All the other studies reported performed semen analysis according to WHO 2010 criteria [72]. (p), progressive motility. #, number.

**Table 4 ijms-22-01988-t004:** Studies reported in Table 2 investigating the relationship between MetS, “unconventional” semen parameters and sex hormones.

Author	Cohort Studied	MetS Definition	# Subjects	Type of Analysis	Sperm DNA Fragmentation	MMP	Testosterone Levels	LH Levels	FSH Levels	Inhibin B Levels
Lotti et al., 2013 * [47]	Males of infertile couples	IDF & AHA/NHLBI	MetS, 27 No-MetS, 324	Comparison of MetS vs. no-MetS men	NE	NE	Lower in MetS men	No difference	No difference	NE
				Correlation between # of MetS components and seminal or hormonal parameters	NE	NE	Negative correlation	No correlation	No correlation	NE
Lotti et al., 2014 [56]	Males of infertile couples	NCEP-ATPIII	MetS, 22 No-MetS, 149	Comparison of MetS vs. no-MetS men	NE	NE	No difference	No difference	No difference	NE
				Correlation between # of MetS components and seminal or hormonal parameters	NE	NE	NE	No correlation	No correlation	NE
Leisegang et al., 2014 * [64]	Men from the general population	IDF & AHA/NHLBI	MetS, 26 No-MetS, 28	Comparison of MetS vs. no-MetS men	Higher in MetS men	Lower in MetS men	Lower in MetS men (saliva)	NE	NE	NE
Leisegang et al., 2016 * [65]	Men from the general population	IDF & AHA/NHLBI	MetS, 42 No-MetS, 32	Comparison of MetS vs. no-MetS men	Higher in MetS men	Lower in MetS men	NE	NE	NE	NE
Ventimiglia et al., 2016 * [57]	Men with primary infertility	NCEP-ATPIII	MetS, 128 No-MetS, 1209	Comparison of MetS vs. no-MetS men	NE	NE	Lower in MetS men	No difference	No difference	Lower in MetS men
Ventimiglia et al., 2017 * [58]	Men with secondary infertility	NCEP-ATPIII	MetS, 20 No-MetS, 147	Comparison of MetS vs. no-MetS men	NE	NE	Lower in MetS men	No difference	No difference	Lower in MetS men
Pilatz et al., 2017 * [67]	27 MetS men and 27 healthy men	IDF	MetS, 27 No-MetS, 27	Comparison of MetS vs. no-MetS men	NE	NE	Lower in MetS men	No difference	No difference	NE
Ehala-Aleksejev and Punab, 2018 * [59]	Males of infertile couples	NCEP-ATPIII	MetS, 471 No-MetS, 2171	Comparison of MetS vs. no-MetS men	NE	NE	Lower in MetS men	Lower in MetS men	No difference	NE
	Fertile men	NCEP-ATPIII	MetS, 29 No-MetS, 209	Comparison of MetS vs. no-MetS men	NE	NE	Lower in MetS men	No difference	No difference	NE
Saikia et al., 2019 * [69]	50 young adult males with MetS and 30 age-matched healthy males	IDF	MetS, 50 No-MetS, 30	Comparison of MetS vs. no-MetS men	NE	NE	Lower in MetS men	NE	Lower in MetS men	Lower in MetS men
Elfassy et al., 2020 [61]	Males of infertile couples	IDF & AHA/NHLBI	MetS, 45 No-MetS, 109	Comparison of MetS vs. no-MetS men	No difference	NE	Lower in MetS men	No difference	No difference	Lower in MetS men
Zhao and Pang, 2020 [70]	Meta-analysis	Various	MetS, 1731 No-MetS, 11740	Comparison of MetS vs. no-MetS men	Higher in MetS men	Lower in MetS men	Lower in MetS men	No difference	Lower in MetS men	Lower in MetS men
Lotti et al., 2020 [63]	248 fertile men	IDF & AHA/NHLBI	MetS, 32 No-MetS, 216	Comparison of MetS vs. no-MetS men	NE	NE	No difference	No difference	No difference	NE
		NCEP-ATPIII	MetS, 17 No-MetS, 231	Comparison of MetS vs. no-MetS men	NE	NE	No difference	No difference	No difference	NE

NE, not evaluated. MMP, mitochondrial membrane potential. * Studies included in Zhao and Pang meta-analysis [70]. In addition: no difference in prolactin levels comparing MetS and no-MetS men has been reported by [47,57,58]; lower AMH levels in MetS vs. no-MetS men have been reported by [57,58]; no difference in estradiol levels comparing MetS and no-MetS men has been reported by [57,58,59,61], while higher estradiol levels in MetS men have been reported by [67]; lower SHBG levels in MetS vs. no-MetS men have been reported by [57,58,61,67]. #, number.

**Table 5 ijms-22-01988-t005:**
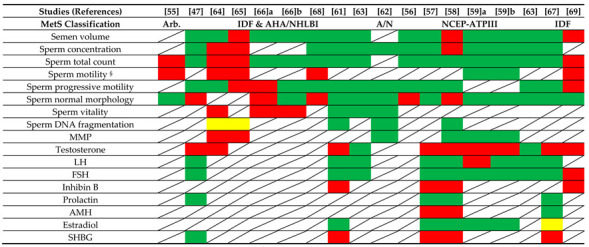
Schematic representation of the associations found in cross-sectional studies between MetS, seminal and hormonal parameters.

The studies have been pooled according to the MetS classification used. Arb, arbitrary (MetS defined as with “hypertension, high LDL cholesterolemia, low HDL cholesterolemia, hyperglycemia, obesity, physical inactivity and blood coagulation disorders”); A/N, AHA/NHLBI classification. § sperm motility refers to sperm “total” motility or “not specified” motility. Rectangle legends: red, lower levels in MetS vs. no-MetS men; yellow, higher levels in MetS vs. no-MetS men; green, no difference between MetS and no-MetS men; white with diagonal, parameter not evaluated. [66]a refers to the evaluation, in [66], of the comparison between infertile men with MetS and fertile men without MetS (see Table 3). [66]b refers to the evaluation, in [66], of the comparison between fertile men with and without MetS (see Table 3). [59]a and [59]b refer to the evaluation, in [59], of the comparison between MetS and no-MetS men in males of infertile couples [59]a and fertile men [59]b (Table 3). [63] compared MetS and no-MetS in fertile men using both IDF & AHA/NHLBI and NCEP-ATPIII classifications.

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
