# Peer review of "Metabolic Syndrome and Reproduction"

_ijms, 2021, doi:10.3390/ijms22041988_

Round 1

Reviewer 1 Report

no remarks - nice work!

The authors presented a comprehensive review concerning an interesting points of the role of metabolic syndrome in male fertility. The manuscript is well-written and will be an interest of the readers.

Author Response

Reviewer 1

We wish to thank the Referee. The comments he/she made were useful and have been incorporated into the final version. Upon the Referee’s suggestions, the manuscript has been edited (see the enclosed copy).

No remarks - nice work!

The authors presented a comprehensive review concerning an interesting points of the role of metabolic syndrome in male fertility. The manuscript is well-written and will be an interest of the readers.

We thank the Referee for the comment.

Reviewer 2 Report

The review entitled “Metabolic syndrome and reproduction” is focused on the prevalence of infertility among male patients affected by metabolic syndrome.

Even if the topic is extremely interesting, the reading of the report is complicated, first of all due to the different definition of metabolic syndrome crossed with different parameters of semen or hormone.

Moreover, the Authors did not report if the analyzed studies are retrospective or prospective (only in 1 case): this represent a very important limitation in the evaluation of a clinical study. I suggest (if possible) to include only perspective studies.

To increase the appeal of the review it would be useful a reorganization of the paragraph according to the analyzed parameters: sperm concentration; motility; morphology; DNA fragmentation, mitochondrial functionality, hormone profile…in order to clearly indicate if the evaluated parameter is affected by the MetS.

In the present form the paragraphs are very long, with a miscellaneous of parameters also according different MetS classification! The results therefore are not very clear to the reader. It would be useful to add a paragraph evaluating the treatment able to increase sperm quality, particularly for metformin widely considered positively affecting male spermatogenesis, both in human and animal HFD models (Metformin Ameliorates Testicular Function and Spermatogenesis in Male Mice with High-Fat and High-Cholesterol Diet-Induced Obesity. Liu CY, Chang TC, Lin SH, Wu ST, Cha TL, Tsao CW. Nutrients. 2020 Jun 29;12(7):1932.  Metformin treatment of high-fat diet-fed obese male mice restores sperm function and fetal growth, without requiring weight loss. McPherson NO, Lane M. Asian J Androl. 2020 Nov-Dec;22(6):560-568. doi: 10.4103/aja.aja_141_19. Association between high-fat diet feeding and male fertility in high reproductive performance mice. Gómez-Elías MD, Rainero Cáceres TS, Giaccagli MM, Guazzone VA, Dalton GN, De Siervi A, Cuasnicú PS, Cohen DJ, Da Ros VG. Sci Rep. 2019 Dec 6;9(1):18546. Metformin improves semen characteristics of oligo-terato-asthenozoospermic men with metabolic syndrome. Morgante G, Tosti C, Orvieto R, Musacchio MC, Piomboni P, De Leo V. Fertil Steril. 2011 May;95(6):2150-2.)

Author Response

Reviewer 2

We wish to thank the Referee. The comments he/she made were useful and have been incorporated into the final version. Upon the Referee’s suggestions, the manuscript has been edited (see the enclosed copy).

The review entitled “Metabolic syndrome and reproduction” is focused on the prevalence of infertility among male patients affected by metabolic syndrome.

Even if the topic is extremely interesting, the reading of the report is complicated, first of all due to the different definition of metabolic syndrome crossed with different parameters of semen or hormone.

We thank the Referee for the comment. In order to make reading easier, in the revised version of the manuscript a summary of the significant associations between MetS and each seminal and hormonal parameter investigated has been provided (page #7, lines 231-233 and pages #12-14, lines 387-470), and a new Table (Table 5) has been added reporting a schematic representation of the results of the studies evaluated. In addition, in Table 5 the studies have been pooled according to the MetS classification used, to facilitate at a glance the evaluation of the associations between MetS, seminal and hormonal parameters investigated in different studies according to each available MetS definition.

Moreover, the Authors did not report if the analyzed studies are retrospective or prospective (only in 1 case): this represent a very important limitation in the evaluation of a clinical study. I suggest (if possible) to include only perspective studies.

We thank the Referee for the comment. As far as we know, there are no longitudinal studies evaluating the relationship between MetS and semen quality. All available studies are cross-sectional in nature, despite the fact that one study (Ozturk et al., 2012) reported to be retrospective, and one (Pilatz et al., 2017) to be prospective. In the new version of the manuscript this point has been clarified (page #8, lines 236-237), and the term “prospective” has been removed (page #9, line 299).

To increase the appeal of the review it would be useful a reorganization of the paragraph according to the analyzed parameters: sperm concentration; motility; morphology; DNA fragmentation, mitochondrial functionality, hormone profile…in order to clearly indicate if the evaluated parameter is affected by the MetS.

We thank the Referee for the comment. According to his/her request, in the revised version of the manuscript a summary of the significant associations between MetS and each seminal and hormonal parameter investigated has been provided (page #7, lines 231-233 and pages #12-14, lines 387-470), and a new Table (Table 5) has been added reporting a schematic representation of the results of the studies evaluated. However, the main text of the old version of the manuscript has been preserved, in order to give an analytical description of the cross-sectional studies evaluated and give further information to the readers in addition to those related to seminal and hormonal parameters (page #7, lines 227-235).

In the present form the paragraphs are very long, with a miscellaneous of parameters also according different MetS classification! The results therefore are not very clear to the reader.

See the comment above.

It would be useful to add a paragraph evaluating the treatment able to increase sperm quality, particularly for metformin widely considered positively affecting male spermatogenesis, both in human and animal HFD models (Metformin Ameliorates Testicular Function and Spermatogenesis in Male Mice with High-Fat and High-Cholesterol Diet-Induced Obesity. Liu CY, Chang TC, Lin SH, Wu ST, Cha TL, Tsao CW. Nutrients. 2020 Jun 29;12(7):1932.  Metformin treatment of high-fat diet-fed obese male mice restores sperm function and fetal growth, without requiring weight loss. McPherson NO, Lane M. Asian J Androl. 2020 Nov-Dec;22(6):560-568. doi: 10.4103/aja.aja_141_19. Association between high-fat diet feeding and male fertility in high reproductive performance mice. Gómez-Elías MD, Rainero Cáceres TS, Giaccagli MM, Guazzone VA, Dalton GN, De Siervi A, Cuasnicú PS, Cohen DJ, Da Ros VG. Sci Rep. 2019 Dec 6;9(1):18546. Metformin improves semen characteristics of oligo-terato-asthenozoospermic men with metabolic syndrome. Morgante G, Tosti C, Orvieto R, Musacchio MC, Piomboni P, De Leo V. Fertil Steril. 2011 May;95(6):2150-2.)

According to the Referee’s request, a paragraph evaluating the treatment of MetS and its impact on semen quality has been added (pages #17-20), focusing on the available studies performed in humans and animal models regarding the effect of hypoglycemic medications (especially metformin), antihypertensive drugs, statins and bariatric surgery on seminal characteristics.

The studies/references suggested by the Referee have been included in the revised manuscript.

Round 2

Reviewer 2 Report

The manuscript has been improved, according to reviewer' suggestions